# Solving Nonconvex-Nonconcave Min-Max Problems exhibiting Weak Minty Solutions

**Axel Böhm**                                                                              *axel.boehm@univie.ac.at*
*University of Vienna,*
*Austria*

**Reviewed on OpenReview:** *https://openreview.net/forum?id=GpOpHyUyrb*

## Abstract

We investigate a structured class of nonconvex-nonconcave min-max problems exhibiting so-called *weak Minty* solutions, a notion which was only recently introduced and already prooved powerful by simultaneously capturing different generalizations of monotonicity. We prove novel convergence results for a generalized version of the optimistic gradient method (OGDA) in this setting, matching the $1/k$ rate for the best iterate in terms of the squared operator norm recently shown for the extragradient method (EG). In addition we propose an adaptive step size version of EG, which does not require knowledge of the problem parameters.

## 1 Introduction

The recent success of machine learning models which can be described by min-max optimization, such as generative adversarial networks (Goodfellow et al., 2014), adversarial learning (Madry et al., 2018), adversarial example games (Bose et al., 2020) or actor-critic methods (Pfau & Vinyals, 2016), has sparked interest in such saddle point problems. While methods have been identified, which (mostly) work in practice, the setting in which the objective function is nonconvex in the minimization and nonconcave in the maximization component remains theoretically poorly understood and even shows intractability results (Daskalakis et al., 2021; Lee & Kim, 2021a). Recently, Daskalakis et al. (2020) studied a class of nonconvex-nonconcave min-max problems and observed that the extragradient method (EG) showed good converge behavior in the experiments. Surprisingly, the problems did not seem to exhibit any of the known tame properties such as monotonicity, or Minty solutions. Later, Diakonikolas et al. (2021) found the appropriate notion (see Assumption 1), which is weaker than the existence of a Minty solution (an assumption extensively used in the literature (Malitsky, 2020; Liu et al., 2020; 2021)) and also generalizes the concept of negative comonotonicity (Bauschke et al., 2020; Combettes & Pennanen, 2004; Lee & Kim, 2021a). Due to these unifying and generalizing properties the notion of weak Minty solutions was promptly studied in (Pethick et al., 2022; Lee & Kim, 2021b).

**Assumption 1** (Weak Minty solution). *Given an operator $F : \mathbb{R}^d \to \mathbb{R}^d$ there exists a point $u^* \in \mathbb{R}^d$ and a parameter $\rho > 0$ such that*

$$\langle F(u), u - u^* \rangle \geq -\frac{\rho}{2}\|F(u)\|^2 \quad \forall u \in \mathbb{R}^d. \tag{1}$$

Additionally, Diakonikolas et al. (2021) proved that a generalization of EG (Korpelevich, 1976) is able to solve problems which exhibit such solutions with a complexity of $\mathcal{O}(\varepsilon^{-1})$ for the squared operator norm. This modification which they title EG+, is based on an aggressive extrapolation step combined with a conservative update step. Such a step size policy has already been explored in the context of a stochastic version of EG in (Hsieh et al., 2020).

In a similar spirit we investigate a variant of the optimistic gradient descent ascent (OGDA) (Daskalakis et al., 2018; Popov, 1980)/Forward-Reflected-Backward (FoRB) (Malitsky & Tam, 2020) method. We pose the question, and give an affirmative answer to:

*Can OGDA match the convergence guarantees of EG in the presence of weak Minty solutions?*

In particular, we show that the following modification of the OGDA method, given for step size $a > 0$ and parameter $0 < \gamma \leq 1$ by

$$u_{k+1} = u_k - a\Big((1+\gamma)F(u_k) - F(u_{k-1})\Big), \quad \forall k \geq 0,$$

is able to match the bounds of EG+ (Diakonikolas et al., 2021; Pethick et al., 2022):

$$u_k = \bar{u}_k - aF(\bar{u}_k), \quad \bar{u}_{k+1} = \bar{u}_k - \gamma aF(u_k), \quad \forall k \geq 0,$$

by only requiring one gradient oracle call per iteration. In Figure 1 we see that beyond the theoretical guarantees OGDA+ can even provide convergence where EG+ does not.

Note that OGDA is most commonly written in the form where $\gamma = 1$, see (Daskalakis et al., 2018; Malitsky & Tam, 2020; Böhm et al., 2022), with the exception of two recent works which have investigated a more general coefficient see (Ryu et al., 2019; Mokhtari et al., 2020). While the previous references target the monotone setting the true importance of $\gamma$ only shows up in the presence of weak Minty solutions as in this case we *require* it to be larger than 1 to guarantee convergence — a phenomenon not present for monotone problems.

**Connection to min-max.** When considering a general (smooth) min-max problem

$$\min_x \max_y f(x, y)$$

the operator $F$ mentioned in Assumption 1 arises naturally as $F(u) := [\nabla_x f(x, y), -\nabla_y f(x, y)]$ with $u = (x, y)$. However, by studying saddle point problems from this more general perspective of variational inequalities (VIs), see (SVI), via the operator $F$ we can simultaneously capture more settings such as certain equilibrium problems, see (Facchinei & Pang, 2007).

**About the weak Minty parameter $\rho$.** The parameter $\rho$ in the definition of weak Minty solutions (1) plays a crucial role in the analysis and the experiments. In particular it is necessary that the step size is larger than a term proportional to $\rho$, see for example Theorem 3.1 or (Pethick et al., 2022). At the same time, as typical, the step size is constrained from above by the reciprocal of the Lipschitz constant of $F$. For example, since the authors of (Diakonikolas et al., 2021) require the step size to be less than $\frac{1}{L}$, their convergence statement only holds if $\rho < \frac{1}{4L}$ for the choice $\gamma = \frac{1}{2}$. This was later improved in (Pethick et al., 2022) to $\frac{1}{L}$ for $\gamma$ even smaller. As in the monotone setting, OGDA however, requires a smaller step size than EG. Nevertheless, through a different analysis we are able to match the most general condition on the weak Minty parameter $\rho < \frac{1}{L}$ for appropriate $\gamma$ and $a$.

**Contribution.**

1. Building on the recently introduced notion of weak solutions to the Minty variational inequality, see (Diakonikolas et al., 2021), we prove a novel convergence rate of $\mathcal{O}(1/k)$ in terms of the squared operator norm for a modification of OGDA, which we name OGDA+, matching the one of EG.

2. Even under the stronger assumption that the operator is moreover monotone we improve the possible range of step sizes for OGDA+ (Ryu et al., 2019) and recover the best known result for the standard method ($\gamma = 1$) (Gorbunov et al., 2022b).

3. We prove a complexity bound of $\mathcal{O}(\varepsilon^{-2})$ for a stochastic version of the OGDA+ method.

4. Additionally, we propose an adaptive step size version of EG+, which is able to obtain the same convergence guarantees without any knowledge of the Lipschitz constant of the operator $F$, and therefore possibly even take larger steps in regions of low curvature and allow for convergence where a fixed step size policy does not.

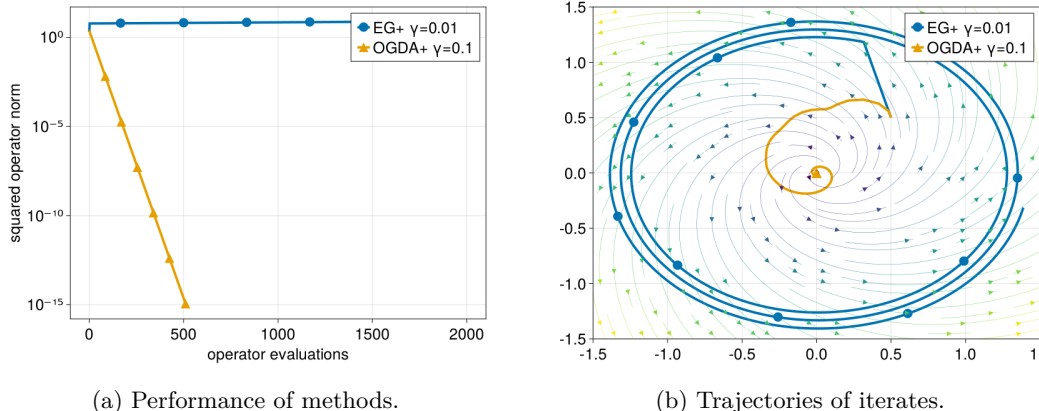

(a) Performance of methods.

(b) Trajectories of iterates.

Figure 1: Problem (7), with $\rho = \frac{1}{L}$, meaning that convergence is not covered by theory. Since the Lipschitz constant can be computed analytically we choose the step size accordingly. Due to the linearity of the operator $F$ there is no benefit in using linesearch, so only methods using constant step sizes are compared. **Only OGDA+ converges**.

## 1.1 Related literature

Since there is an extensive literature on convergence rates in terms of a gap function or distance to a solution for monotone problems as well as generalizations such as nonconvex-concave (Boţ & Böhm, 2020; Lin et al., 2020), convex-nonconcave (Xu et al., 2023) or under the Polyak-Łojasiewicz assumption, see (Yang et al., 2020), we will only focus on the nonconvex-*non*concave setting.

**Weak minty.** Diakonikolas et al. (2021) noticed that a particular parametrization of the von Neumann ratio game exhibits a new type of solution, which they titled weak Minty, without possessing any of the known properties such as (negative) comonotonicity or Minty solutions. They showed convergence in the presence of such solutions for EG if the extrapolation step size is twice as large as the update step. Later Pethick et al. (2022) showed that the condition on the weak Minty parameter can be relaxed by reducing the length of the update step even further and they do so in an adaptive way. In order to not require any other hyperparameters they also propose a backtracking line search, which might come at the cost of additional gradient computations or the use of second order information (in contrast to the adaptive step size we propose in Algorithm 3). In (Lee & Kim, 2021b) a different approach is taken by restricting the attention to the min-max setting and using multiple ascent steps per descent step, obtaining the same $\mathcal{O}(1/k)$ rate as EG.

**Minty solutions.** Many works have shown different approaches for when the problem at hand exhibits a Minty solution, see (MVI). The authors of (Liu et al., 2021) showed that weakly monotone VIs can be solved by successively adding a quadratic proximity term and repeatedly optimizing the resulting strongly monotone VI with any convergent method. In (Mertikopoulos et al., 2019) the convergence of the OGDA method was proven, but without any rate. In (Malitsky, 2020) it was noted that the convergence proof for the golden ratio algorithm (GRAAL) works without any modification. See also (Dang & Lan, 2015) for a non-euclidean version of EG and (Liu et al., 2020) for adaptive methods. While the assumption that a Minty solution exists is a generalization of the monotone setting it is difficult to find nonmonotone problems that do possess such solutions. In our setting, see Assumption 1, the Minty inequality (MVI) is allowed to be violated at every point by a factor proportional to the squared operator norm.

**Negative comonotonicity.** While previously studied under the name of *cohypomonotonicity* (Combettes & Pennanen, 2004) the notion of negative comonotonicity was recently explored in (Bauschke et al., 2020). It provides a generalization of monotonicity, but in a direction different from the notion of Minty solutions and only a few works have analyzed methods in this setting. The authors of (Lee & Kim, 2021a) studied an anchored version of EG and showed an improved convergence rate of $\mathcal{O}(1/k^2)$ (in terms of the squared operator norm). Similarly, Cai et al. (2022) studied an accelerated version of the reflected gradient method

(Malitsky, 2015). It is an open question whether such an acceleration is possible in the more general setting of weak Minty solutions (any Stampacchia solution to the VI given by negatively comonotone operator is a weak Minty solution). Another interesting observation was made in (Gorbunov et al., 2022c) where for cohypomonotone problems monotonically decreasing gradient norm was shown when using EG. However, we did not observe this in our experiments, highlighting the need to distinguish this class from problems with weak Minty solutions.

**Interaction dominance.** The authors of (Grimmer et al., 2022) investigate the notion of *α-interaction dominance* for nonconvex-nonconcave min-max problems and showed that the proximal-point method converges sublinearly if this condition holds in $y$ and linearly if it holds in both components. Furthermore Lee & Kim (2021a) showed that if a problem is interaction dominant in both components, then it is also negatively comonotone.

**Optimism.** The beneficial effects of introducing the simple modification commonly known as optimism have recently sparked the interest of the machine learning community (Daskalakis et al., 2018; Daskalakis & Panageas, 2018; Liang & Stokes, 2019; Gidel et al., 2019). Its name originates from online optimization (Rakhlin & Sridharan, 2013a;b). The idea dates back even further (Popov, 1980) and has been studied in the mathematical programming community as well (Malitsky, 2015; Malitsky & Tam, 2020; Csetnek et al., 2019).

## 2 Preliminaries

### 2.1 Notions of solution

We summarize the most commonly used notions of solutions appearing in the context of variational inequalities (VIs) and beyond. Note that these are commonly defined in terms of a constraint set $C \subset \mathbb{R}^d$. A *Stampacchia*[1] (Kinderlehrer & Stampacchia, 2000) solution of the VI given by $F : \mathbb{R}^d \to \mathbb{R}^d$ is defined as a point $u^*$ such that

$$\langle F(u^*), u - u^* \rangle \geq 0 \quad \forall u \in C. \tag{SVI}$$

In particular we only consider in this manuscript the unconstrained case $C = \mathbb{R}^d$ in which case the above condition reduces to $F(u^*) = 0$. Very much related with a long tradition is the following. A *Minty*[1] solution is given by a point $u^* \in C$ such that

$$\langle F(u), u - u^* \rangle \geq 0 \quad \forall u \in C. \tag{MVI}$$

If $F$ is continuous, a Minty solution of the VI is always a Stampacchia solution. The reverse is in general not true but holds for example if the operator $F$ is monotone. In particular, there exist nonmonotone problems with Stampacchia solutions but without any Minty solutions.

### 2.2 Notions of monotonicity

The aim of this section is to recall some elementary and some more recent notions of monotonicity and the connection between those. We call an operator $F$ **monotone** if

$$\langle F(u) - F(v), u - v \rangle \geq 0.$$

Such operators arise naturally as the gradients of convex functions, from convex-concave min-max problems or from equilibrium problems.

Two notions frequently studied that fall in this class are **strongly monotone** operators fulfilling

$$\langle F(u) - F(v), u - v \rangle \geq \mu \|u - v\|^2.$$

---

[1]Sometimes Stampacchia and Minty solutions are referred to as *strong* and *weak* solutions respectively, see (Nesterov, 2007), but we will refrain from this nomenclature, as it is confusing in the context of *weak Minty* solutions.

They appear as gradients of strongly convex functions or strongly-convex-strongly-concave min-max problems. A second subclass of monotone operators are so-called **cocoercive** operators fulfilling

$$\langle F(u) - F(v), u - v \rangle \geq \beta \| F(u) - F(v) \|^2. \tag{2}$$

They appear, for example, as gradients of smooth convex functions, in which case (2) holds with $\beta$ equal to the reciprocal of the gradients Lipschitz constant.

**Leaving the monotone world.** Both subclasses of monotonicity introduced above can be used as starting points to venture into the non-monotone world. Since general non-monotone operators might exhibit erratic behavior like periodic cycles and spurious attractors (Hsieh et al., 2021), it makes sense to find settings that extend the monotone one, but still remain tractable. First and foremost, the by now well-studied setting of $\nu$-**weak monotonicity**

$$\langle F(u) - F(v), u - v \rangle \geq -\nu \| u - v \|^2.$$

Such operators arise as the gradients of the well-studied class (see (Davis & Drusvyatskiy, 2018)) of weakly convex functions — a rather generic class of functions as it includes all functions *without upward cusps*. In particular every smooth function with Lipschitz gradient turns out to fulfill this property. On the other hand, extending the notion of cocoercivity to allow for negative coefficients, referred to as **cohypomonotonicity**, has received much less attention (Bauschke et al., 2020; Combettes & Pennanen, 2004) and is given by

$$\langle F(u) - F(v), u - v \rangle \geq -\gamma \| F(u) - F(v) \|^2.$$

Clearly, if there exists a Stampacchia solution for such an operator, then it also fulfills Assumption 1.

**Behavior w.r.t. the solution.** While the above properties are standard assumption in the literature, it is usually sufficient to ask for the corresponding condition to hold when one of the arguments is a (Stampacchia) solution. This means instead of monotonicity it is enough to obtain standard convergence results, see (Gorbunov et al., 2022a), to ask for the operator $F$ to be **star-monotone** (Pennanen, 2001), i.e.

$$\langle F(u), u - u^* \rangle \geq 0$$

or **star-cocoercive** (Gorbunov et al., 2022a)

$$\langle F(u), u - u^* \rangle \geq \gamma \| F(u) \|^2.$$

In this spirit, we can provide a new interpretation to the assumption of the existence of a weak Minty solution as asking for the operator $F$ to be **negatively star-cocoercive** (with respect to at least one solution). Furthermore, we want to point out that while the above star notions are sometimes required to hold for all solutions $u^*$, in the following we only require it to hold for a single solution.

## 3 OGDA for problems with weak Minty solutions

The generalized version of OGDA, whose name we equip in the spirit of (Diakonikolas et al., 2021; Pethick et al., 2022), with a "+" to highlight the presence of the additional parameter $\gamma$, is given by:

---
**Algorithm 1** OGDA+
---
**Require:** Starting point $u_0 = u_{-1} \in \mathbb{R}^d$, step size $a > 0$ and parameter $0 < \gamma \leq 1$.
   **for** $k = 0, 1, \ldots$ **do**
      $u_{k+1} = u_k - a\Big( (1 + \gamma) F(u_k) - F(u_{k-1}) \Big)$

---

**Theorem 3.1.** *Let $F : \mathbb{R}^d \to \mathbb{R}^d$ be $L$-Lipschitz continuous satisfying Assumption 1 with $\frac{1}{L} > \rho$ and $(u_k)_{k \geq 0}$ be the iterates generated by Algorithm 1 with step size $a$ satisfying $a > \rho$ and*

$$aL \leq \frac{1 - \gamma}{1 + \gamma}. \tag{3}$$

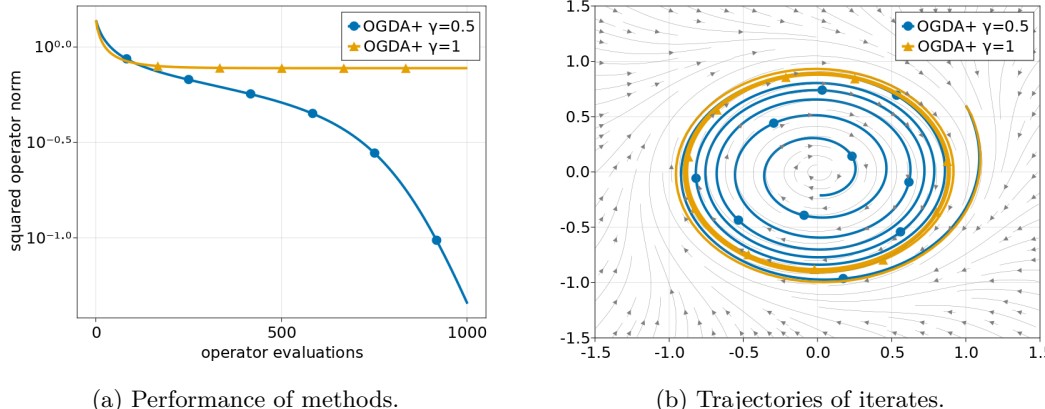

(a) Performance of methods.

(b) Trajectories of iterates.

Figure 2: **Polar game** from (Pethick et al., 2022) with $\frac{1}{8L} > \rho > \frac{1}{10L}$. Shows the need to reduce $\gamma$ in OGDA+.

*Then, for all $k \geq 0$*

$$\min_{i=0,\ldots,k-1} \|F(u_i)\|^2 \leq \frac{1}{ka\gamma(a-\rho)} \|u_0 + aF(u_0) - u^*\|^2.$$

*In particular as long as $\rho < \frac{1}{L}$ we can find a $\gamma$ small enough such that the above bound holds.*

The first observation is that we would like to choose $a$ as large as possible as this allows us to treat the largest class of problems with $\rho < a$. In order to be able to choose the step size $a$ large we have to decrease $\gamma$ as evident from (3). This, however degrades the speed of the algorithm as it makes the update steps smaller — the same effect can be observed (Pethick et al., 2022) for EG+ and is therefore not surprising. One could derive an optimal $\gamma$ (i.e. minimizing the right hand side) from Theorem 3.1, which however results in a uninsightful cubic dependence on $\rho$. In practice, the strategy of decreasing $\gamma$ until we get convergence, but not further gives reasonable results.

Furthermore, we want to point out that the condition $\rho < \frac{1}{L}$, is precisely the best possible bound for EG+ in (Pethick et al., 2022).

### 3.1 Improved bounds under monotonicity

While the above theorem also holds if the operator $F$ is monotone, we can modify the proof slightly to obtain a better dependence on the parameters:

**Theorem 3.2.** *Let $F : \mathbb{R}^d \to \mathbb{R}^d$ be monotone and $L$-Lipschitz. If $aL = \frac{2-\gamma}{2+\gamma} - \varepsilon$ for $\varepsilon > 0$ then, the iterates generated by OGDA+ fulfill*

$$\min_{i=0,\ldots,k-1} \|F(u_i)\|^2 \leq \frac{2}{ka^2\gamma^2\varepsilon} \|u_0 + aF(u_0) - u^*\|^2.$$

*In particular, we can choose $\gamma = 1$ and $a < \frac{1}{3L}$.*

There are different works discussing the convergence of OGDA in terms of the iterates or a gap function with $a < \frac{1}{2L}$, see for example (Malitsky & Tam, 2020). We, however, want to compare the above bound to more similar results on rates for the best iterate in terms of the operator norm. The same rate as ours for OGDA is shown in (Chavdarova et al., 2021) but requires the conservative step size bound $a \leq \frac{1}{16L}$. This was later improved to $a \leq \frac{1}{3L}$ in (Gorbunov et al., 2022b) where the bound even holds for the last iterate. However, all of these only deal with the case $\gamma = 1$. The only other reference that deals with a generalized (i.e. not necessarily $\gamma = 1$) version of OGDA is (Ryu et al., 2019). There the resulting step size condition is $a \leq \frac{2-\gamma}{4L}$, which is strictly worse than ours for any $\gamma$. To summarize, not only do we show for the first time that the step size of a generalization of OGDA can go above $\frac{1}{2L}$ we also provide the least restrictive bound for any value of $\gamma$.

### 3.2 OGDA+ stochastic

In this section we discuss the setting where instead of the exact operator $F$ we only have access to a collection of independent estimators $\tilde{F}(\cdot, \xi_i)$ at every iteration. We assume here that estimator $\tilde{F}$ is unbiased $\mathbb{E}\big[\tilde{F}(u_k, \xi_i) \,\big|\, u_{k-1}\big] = F(u_k)$, and has bounded variance $\mathbb{E}\big[\|\tilde{F}(u_k, \xi_i) - F(u_k)\|^2\big] \leq \sigma^2$, we show that we can still guarantee convergence, by using batch sizes $B$ of order $\mathcal{O}(\varepsilon^{-1})$:

---

**Algorithm 2** stochastic OGDA+

---

**Require:** Starting point $u_0 = u_{-1} \in \mathbb{R}^d$, step size $a > 0$, parameter $0 < \gamma \leq 1$ and batch size $B$.

    **for** $k = 0, 1, \dots$ **do**

        Sample i.i.d. $(\xi_i)_{i=1}^B$ and compute estimator $\tilde{g}_k = \frac{1}{B} \sum_{i=1}^B \tilde{F}(u_k, \xi_i)$

        $u_{k+1} = u_k - a\Big( (1 + \gamma)\, \tilde{g}_k - \tilde{g}_{k-1} \Big)$

---

**Theorem 3.3.** *Let $F : \mathbb{R}^d \to \mathbb{R}^d$ be $L$-Lipschitz satisfying Assumption 1 with $\frac{1}{L} > \rho$ and let $(u_k)_{k \geq 0}$ be the sequence of iterates generated by stochastic OGDA+, with $a$ and $\gamma$ satisfying $\rho < a < \frac{1-\gamma}{1+\gamma}\frac{1}{L}$ then, to visit an $\varepsilon$ stationary point such that $\min_{i=0,\dots,k-1} \mathbb{E}\big[\|\tilde{F}(u_i, \xi)\|^2\big] \leq \varepsilon$ we require*

$$\mathcal{O}\left( \frac{1}{\varepsilon} \frac{1}{a\gamma(a - \rho)} \mathbb{E}\big[\|u_0 + a\tilde{g}_0 - u^*\|^2\big] \max\left\{ 1, \frac{4\sigma^2}{aL}\frac{1}{\varepsilon} \right\} \right)$$

*calls to the stochastic oracle $\tilde{F}$, with large batch sizes of order $\mathcal{O}(\varepsilon^{-1})$.*

In practice large batch sizes of order $\mathcal{O}(\varepsilon^{-1})$ are typically not desirable, but rather a small or decreasing step size is preferred. In the weak Minty setting this is cause for additional trouble due to the necessity of large step sizes to guarantee convergence. See in this context the heuristic variant proposed in (Pethick et al., 2022) which decreases the parameter corresponding to $\gamma$ in our setting. Unfortunately the current analysis does not allow for variable $\gamma$.

## 4 EG+ with adaptive step sizes

In this section we present Algorithm 3 that is able to solve the previously mentioned problems without any knowledge of the Lipschitz constant $L$, as it is typically difficult to compute in practice. Additionally, it is well known that rough estimates will lead to small step sizes and slow convergence behavior. However, in the presence of weak Minty solutions there is additional interest in choosing large step sizes. We observed in Theorem 3.1 and related works such as (Diakonikolas et al., 2021) and (Pethick et al., 2022) the fact that a crucial ingredient in the analysis is that the step size is chosen larger than a multiple of the weak Minty parameter $\rho$ to guarantee convergence at all. For these reasons we want to outline a method using adaptive step sizes, meaning that no step size needs to be supplied by the user and no line-search is carried out.

Since the analysis of OGDA+ is already quite involved in the constant step size regime we choose to equip EG+ with an *adaptive step size* which estimates the inverse of the (local) Lipschitz constant, see (4). Due the fact that the literature on adaptive methods, especially in the context of VIs is so vast we do not aim to give a comprehensive review but highlight only few with especially interesting properties. In particular we do not want to touch on methods with linesearch procedure which typically result in multiple gradient computations per iteration, such as (Tseng, 2000; Malitsky & Tam, 2020).

We use a simple and therefore widely used step size choices which naively estimates the local Lipschitz constant and forces a monotone decreasing behavior. Such step sizes have been used extensively for monotone VIs, see (Yang & Liu, 2018; Boţ et al., 2020), and similarly in the context of the mirror-prox method which corresponds to EG in the setting of (non-euclidean) Bregman distances, see (Antonakopoulos et al., 2019).

A version of EG with a different adaptive step size choice has been investigated by Antonakopoulos et al. (2021); Bach & Levy (2019) with the unique feature that it is able to achieve the optimal rates for both smooth and nonsmooth problems without modification. However, these rates are only for monotone VIs and are in terms of the gap function.

One of the drawbacks of adaptive methods resides in the fact that the step sizes are typically required to be nonincreasing which results in poor behavior if a high curvature area was visited by the iterates before reaching a low curvature region. To the best of our knowledge the only method which is allowed to use nonmonotone step size to treat VIs, and does not use a possibly costly linesearch, is the golden ratio algorithm (Malitsky, 2020). It comes with the additional benefit of not requiring a global bound on the Lipschitz constant of $F$ at all. While it is known that this method converges under the stronger assumption of existence of Minty solutions, a quantitative convergence result is still open.

---

**Algorithm 3** EG+ with adaptive step size

---

**Require:** Starting points $u_0, \bar{u}_0 \in \mathbb{R}^d$, initial step size $a_0$ and parameter $\tau \in (0,1)$ and $0 < \gamma \le 1$.

1: **for** $k = 0, 1, \dots$ **do**

2:     Find the step size:

$$a_k = \min \left\{ a_{k-1}, \frac{\tau \|u_{k-1} - \bar{u}_{k-1}\|}{\|F(u_{k-1}) - F(\bar{u}_{k-1})\|} \right\}. \tag{4}$$

3:     Compute next iterate:

$$u_k = \bar{u}_k - a_k F(\bar{u}_k)$$
$$\bar{u}_{k+1} = \bar{u}_k - a_k \gamma F(u_k).$$

---

Clearly, $a_k$ is monotonically decreasing by construction. Moreover, it is bounded away from zero by the simple observation that $a_k \ge \min\{a_0, \tau/L\} > 0$. The sequence therefore converges to a positive number which we denote by $a_\infty := \lim_k a_k$.

**Theorem 4.1.** *Let $F : \mathbb{R}^d \to \mathbb{R}^d$ be $L$-Lipschitz that satisfies Assumption 1, where $u^*$ denotes any weak Minty solution, with $a_\infty > 2\rho$ and let $(u_k)_{k \ge 0}$ be the iterates generated by Algorithm 3 with $\gamma = \frac{1}{2}$ and $\tau \in (0,1)$. Then, there exists a $k_0 \in \mathbb{N}$, such that*

$$\min_{i=k_0,\dots,k} \|F(u_k)\|^2 \le \frac{1}{k - k_0} \frac{L}{\tau(\frac{a_\infty}{2} - \rho)} \|\bar{u}_{k_0} - u^*\|^2.$$

Algorithm 3 presented above provides several benefits, but also some drawbacks. The main advantage resides in the fact that the Lipschitz constant of the operator $F$ does not need to be known. Moreover, the step size choice presented in (4) might allow us to take steps much larger than what would be suggested by a global Lipschitz constant if the iterates never — or only during later iterations — visits the region of high curvature (large local $L$). In such cases these larger step sizes come with the additional advantage that they allow us to solve a richer class of problems as we are able to relax the condition $\rho < \frac{1}{4L}$ in the case of EG+ to $\rho < a_\infty/2$ where $a_\infty = \lim_k a_k \ge \tau/L$.

On the other hand, we face the problem that the bounds in Theorem 4.1 only hold after an unknown number of initial iterations when $a_k/a_{k+1} \le \frac{1}{\tau}$ is finally satisfied. In theory this might take long if the curvature around the solution is much higher than in the starting area as this will force the need to decrease the step size very late into the solution process resulting in the quotient $a_k/a_{k+1}$ being too large. This drawback could be mitigated by choosing $\tau$ smaller. However, this will result in poor performance due to small step sizes. Even for monotone problems where this type of step size has been proposed this problem could not be circumvented and authors instead focused on convergence of the iterates without any rate.

## 5 Numerical experiments

In the following we compare EG+ method from (Diakonikolas et al., 2021) with the two methods we propose OGDA+ and EG+ with adaptive step size, see Algorithm 1 and Algorithm 3 respectively. Last but not least we also include the CurvatureEG+ method from (Pethick et al., 2022) which is a modification of EG+ and adaptively chooses the ratio of extrapolation and update step. In addition a backtracking linesearch

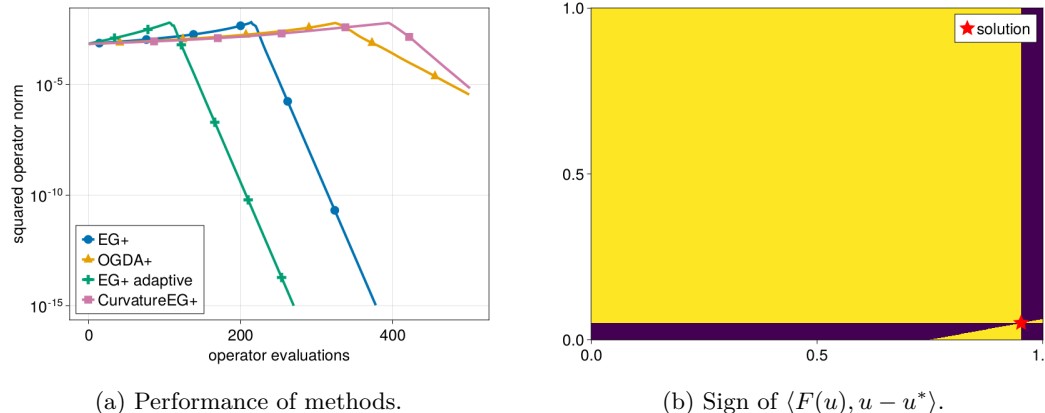

(a) Performance of methods.      (b) Sign of $\langle F(u), u - u^* \rangle$.

Figure 3: **Ratio game**. A particularly difficult parametrization of (5) suggested in (Daskalakis et al., 2020). Right: The sign of $\langle F(u), u - u^* \rangle$ parametrized as $u := (x, 1 - x, y, 1 - y)$, with yellow representing a negative sign and purple a positive one, highlighting the fact that the solution is not Minty. Nevertheless, all methods are able to converge.

is performed with an initial guess made by second order information, whose extra cost we ignore in the experiments.

## 5.1 Von Neumann's ratio game

We consider von Neumann's *ratio game* (von Neumann, 1945) recently explored in (Daskalakis et al., 2020; Diakonikolas et al., 2021). It is given by

$$\min_{x \in \Delta^m} \max_{y \in \Delta^n} V(x, y) = \frac{\langle x, Ry \rangle}{\langle x, Sy \rangle}, \tag{5}$$

where $R \in \mathbb{R}^{m \times n}$ and $S \in \mathbb{R}_+^{m \times n}$ with $\langle x, Sy \rangle > 0$ for all $x \in \Delta^m, y \in \Delta^n$, with $\Delta^d := \{z \in \mathbb{R}^d : z_i \geq 0, \sum_{i=1}^d z_i = 1\}$ denoting the unit simplex. Expression (5) can be interpreted as the value $V(\pi_x, \pi_y)$ for a stochastic game with a single state and mixed strategies.

In Figure 3b we see an illustration of a particularly difficult instance of (5) discussed in (Daskalakis et al., 2020), highlighting the fact that the Stampacchia solution is not a Minty solution, even when restricted to arbitrarily close ball around it (yellow area touching the solution). Interestingly we still observe good convergence behavior, although an estimated $\rho$ is more than ten times larger than the estimated Lipschitz constant.

## 5.2 Forsaken

A particularly difficult min-max toy example with "Forsaken" solution was proposed in Example 5.2 of (Hsieh et al., 2021), and is given by

$$\min_{x \in \mathbb{R}} \max_{y \in \mathbb{R}} x(y - 0.45) + \varphi(x) - \varphi(y), \tag{6}$$

where $\varphi(z) = \frac{1}{4}z^2 - \frac{1}{2}z^4 + \frac{1}{6}z^6$. This problem exhibits a Stampacchia solution at $(x^*, y^*) \approx (0.08, 0.4)$, but also *two* limit cycles not containing any critical point of the objective function. In addition, Hsieh et al. (2021) also observed that the limit cycle closer to the solution repels possible trajectories of iterates, thus "shielding" the solution. Later, Pethick et al. (2022) noticed that, restricted to the box $\|(x, y)\|_\infty < \frac{3}{2}$ the above mentioned solution is weak Minty with $\rho \geq 2 \cdot 0.477761$, which is much larger than $\frac{1}{L} \approx 0.08$. In line with these observations we can see in Figure 4 that none of the fixed step size methods with step size bounded by $\frac{1}{L}$ converge. In light of this observation Pethick et al. (2022) proposed a backtracking linesearch which potentially allows for larger steps than predicted by the global Lipschitz constant. Similarly, our proposed

adaptive step size version of EG+, see Algorithm 3, is also able to break through the repelling limit cycle and converge the solution. On top of this, it does so at a faster rate and without the need of additional computations in the backtracking procedure.

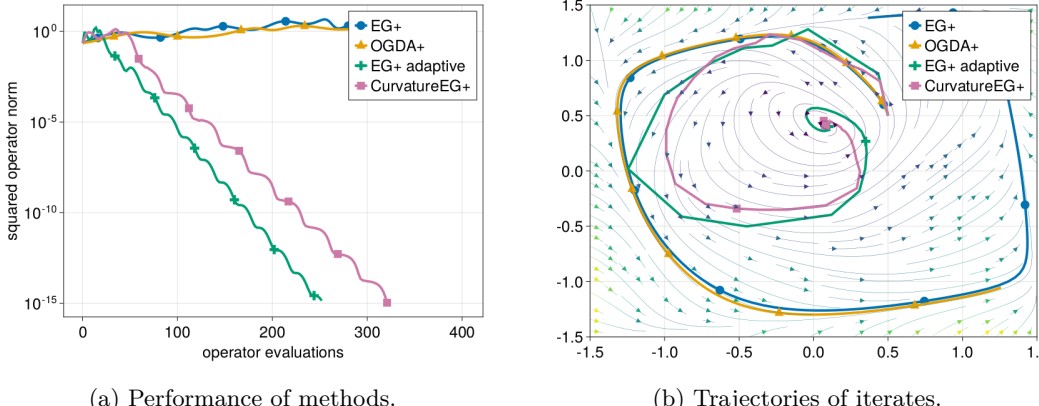

(a) Performance of methods.  (b) Trajectories of iterates.

Figure 4: **Forsaken.** An illustration of different methods on problem (6), originally suggested by Hsieh et al. (2021). Only Algorithm 3 and CurvatureEG+ are able to choose a step size large enough to withstand the repellent limit cycle.

## 5.3 Lower bound example

The following min-max problem was introduced in (Pethick et al., 2022) as a lower bound on the dependence between $\rho$ and $L$ for EG+:

$$\min_{x \in \mathbb{R}} \max_{y \in \mathbb{R}} \xi xy + \frac{\zeta}{2}(x^2 - y^2). \tag{7}$$

In particular Theorem 3.4 from (Pethick et al., 2022) states that EG+ (with any $\gamma$) and constant step size $a = \frac{1}{L}$ converges for this problem if and only if $(0,0)$ is a weak Minty solution with $\rho < \frac{1-\gamma}{L}$, where $\rho$ and $L$ can be computed explicitly in the above example and are given by

$$L = \sqrt{\xi^2 + \zeta^2} \quad \text{and} \quad \rho = -2\frac{\zeta}{\xi^2 + \zeta^2}.$$

Figure 1 is obtained by choosing $\xi = \sqrt{3}$ and $\zeta = -1$ we get exactly $\rho = \frac{1}{L}$ and the theory therefore predicting divergence of EG+ for any $\gamma$, which is exactly what is empirically observed. Although, the general upper bound proved in Theorem 3.1 only states convergence in the case $\rho < \frac{1}{L}$, we observe rapid convergence of OGDA+ for this example showcasing that it can drastically outperform EG+ in some scenarios.

## 6 Conclusion

Many interesting questions remain in the realm of min-max problems — especially when leaving the convex-concave setting. Very recently Gorbunov et al. (2022c) showed that the $\mathcal{O}(1/k)$ bounds on the squared operator norm for EG and OGDA for the **last iterate** (and not just the best one) hold even in the negative comonotone setting. Deriving a similar statement in the presence of merely weak Minty solutions is an open question.

Overall, our analysis and experiments seem to provide evidence that there is little advantage of using OGDA+ over EG+ for most problems as the lower iteration cost is offset by the smaller step size. One exception is given by problem (7) displayed in Figure 1, which is not covered by theory and OGDA+ is the only method able to converge.

Lastly, we observe that the previous paradigm in pure minimization of "smaller step size ensures convergence" but "larger step size gets there faster", where the latter is typically constrained by the reciprocal of the

gradients Lipschitz constant, does not seem to hold true for min-max problems anymore. The analysis of different methods in the presence of weak Minty solutions shows that convergence can be lost if the step size is too small and sometimes needs to be larger than $\frac{1}{L}$, which one can typically only hope for in adaptive methods. Our EG+ method with adaptive step size achieves this even without the additional cost of a backtracking linesearch as used for the CurvatureEG+ method of (Pethick et al., 2022).

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

# A  Omitted proofs

## A.1  OGDA+

For convenience we will sometimes use the notation $g_k$ for $F(u_k)$ for all $k \geq -1$.

**Lemma A.1.** *Let $(u_k)$ be the sequence of iterates generated by Algorithm 1, then*

$$\|u_{k+1} + ag_k - u^*\|^2 \leq \|u_k + ag_{k-1} - u^*\|^2 + a\gamma\rho\|g_k\|^2 + 4a\gamma^{-1}\langle g_k - g_{k-1}, u_k - u_{k+1}\rangle \\ - (2\gamma^{-1} - 1)\|u_{k+1} - u_k\|^2 - a^2(1 + 2\gamma^{-1})\|g_k - g_{k-1}\|^2. \tag{8}$$

*Proof.* From the update of the method we deduce for all $k \geq 0$

$$\begin{aligned} \|u_{k+1} + ag_k - u^*\|^2 &= \|u_k - a\gamma g_k + ag_{k-1} - u^*\|^2 \\ &= \|u_k + ag_{k-1} - u^*\|^2 - 2\langle u_k - u^*, a\gamma g_k\rangle - \langle 2ag_{k-1} - a\gamma g_k, a\gamma g_k\rangle \\ &\leq \|u_k + ag_{k-1} - u^*\|^2 + a\gamma\rho\|g_k\|^2 - \langle 2ag_{k-1} - a\gamma g_k, a\gamma g_k\rangle, \end{aligned} \tag{9}$$

where we used the weak Minty assumption to deduce the last inequality. It remains to derive the following equality

$$-\langle 2ag_{k-1} - a\gamma g_k, a\gamma g_k\rangle = 4a\gamma^{-1}\langle g_k - g_{k-1}, u_k - u_{k+1}\rangle - (2\gamma^{-1} - 1)\|u_{k+1} - u_k\|^2 \\ - a^2(1 + 2\gamma^{-1})\|g_k - g_{k-1}\|^2. \tag{10}$$

This can be seen by expressing every difference of iterates in terms of gradients according to Algorithm 1, giving

$$\begin{aligned} 4a\gamma^{-1}\langle g_k - g_{k-1}, u_k - u_{k+1}\rangle &= 4a\gamma^{-1}\langle g_k - g_{k-1}, a(g_k - g_{k-1}) + a\gamma g_k\rangle \\ &= 4a^2\gamma^{-1}\|g_k - g_{k-1}\|^2 + 4a^2\langle g_k, g_k - g_{k-1}\rangle \end{aligned}$$

and

$$
\begin{aligned}
&- (2\gamma^{-1} - 1)\|u_{k+1} - u_k\|^2 \\
&= (1 - 2\gamma^{-1})\|a(g_k - g_{k-1}) + a\gamma g_k\|^2 \\
&= (a^2 - 2a^2\gamma^{-1})\|g_k - g_{k-1}\|^2 + (2\gamma a^2 - 4a^2)\langle g_k, g_k - g_{k-1}\rangle + (a^2\gamma^2 - 2a^2\gamma)\|g_k\|^2 \\
&= (a^2 - 2a^2\gamma^{-1})\|g_k - g_{k-1}\|^2 + \langle g_k, (a^2\gamma^2 - 4a^2)g_k - (2\gamma a^2 - 4a^2)g_{k-1}\rangle.
\end{aligned}
$$

Adding the previous two equalities to $-a^2(1+2\gamma^{-1})\|g_k - g_{k-1}\|^2$ proves (10). Combining (9) and (10) proves the desired statement. $\qquad\square$

We are actually going to show a slightly more general version of Theorem 3.1, which introduces an additional parameter $\lambda$. Note that for $\lambda = \gamma^{-1}$ we recover the statement of Theorem 3.1. This allows us to cover the analysis of the monotone case in one proof. In particular $\lambda$ close to zero will yield the statement of Theorem 3.2.

**Theorem A.1.** *Let $F : \mathbb{R}^d \to \mathbb{R}^d$ be L-Lipschitz continuous satisfying Assumption 1, where $u^*$ denotes any weak Minty solution, with $a\lambda\gamma > \rho$ for some $0 \leq \lambda \leq 2\gamma^{-1}$ and let $(u_k)_{k\geq 0}$ be the sequence of iterates generated by Algorithm 1 with*

$$
aL \leq \frac{2 - \lambda\gamma - \gamma}{2 - \lambda\gamma + \gamma}. \tag{11}
$$

*Then, for all $k \geq 0$*

$$
\frac{1}{k}\sum_{i=0}^{k-1}\|F(u_i)\|^2 \leq \frac{1}{ka\gamma(a\lambda\gamma - \rho)}\|u_0 + aF(u_0) - u^*\|^2.
$$

*In particular as long as $\rho < \frac{1}{L}$ we can find a small enough $\gamma$ such that the above bound holds.*

*Proof.* Using the definition $u_{k+1}$ we can express

$$
a\gamma g_k = u_k - u_{k+1} - a\lambda(g_k - g_{k-1}). \tag{12}
$$

By applying norms on both sides we obtain via expansion of squares

$$
a^2\lambda\gamma^2\|g_k\|^2 = \lambda\|u_{k+1} - u_k\|^2 + a^2\lambda\|g_k - g_{k-1}\|^2 + 2a\lambda\langle u_{k+1} - u_k, g_k - g_{k-1}\rangle. \tag{13}
$$

Adding (13) and Lemma A.1 we deduce

$$
\begin{aligned}
\|u_{k+1} + ag_k - u^*\|^2 + a\gamma(a\lambda\gamma - \rho)\|g_k\|^2 &\leq \|u_k + ag_{k-1} - u^*\|^2 - a^2(1 + 2\gamma^{-1} - \lambda)\|g_k - g_{k-1}\|^2 \\
&\quad - (2\gamma^{-1} - 1 - \lambda)\|u_{k+1} - u_k\|^2 + (4a\gamma^{-1} - 2a\lambda)\langle u_k - u_{k+1}, g_k - g_{k-1}\rangle. \tag{14}
\end{aligned}
$$

Now, we get via Young's inequality

$$
(4\gamma^{-1}a - 2a\lambda)\langle u_k - u_{k+1}, g_k - g_{k-1}\rangle \leq aL(2\gamma^{-1} - \lambda)\|u_k - u_{k+1}\|^2 + L^{-1}a(2\gamma^{-1} - \lambda)\|g_k - g_{k-1}\|^2.
$$

Combining the previous two inequalities

$$
\begin{aligned}
\|u_{k+1} + ag_k - u^*\|^2 + a\gamma(a\gamma\lambda - \rho)\|g_k\|^2 &\leq \|u_k + ag_{k-1} - u^*\|^2 \\
&+ (aL^{-1}(2\gamma^{-1} - \lambda) - a^2(1 + 2\gamma^{-1} - \lambda))\|g_k - g_{k-1}\|^2 - (2\gamma^{-1} - 1 - \lambda - La(2\gamma^{-1} - \lambda))\|u_{k+1} - u_k\|^2. \tag{15}
\end{aligned}
$$

We now use the Lipschitz continuity[2] of $F$ to deduce that

$$
\begin{aligned}
\|u_{k+1} + ag_k - u^*\|^2 + a\gamma(a\gamma\lambda - \rho)\|g_k\|^2 &\leq \|u_k + ag_{k-1} - u^*\|^2 \\
&+ (aL(2\gamma^{-1} - \lambda) - a^2L^2(1 + 2\gamma^{-1} - \lambda))\|u_k - u_{k-1}\|^2 - (2\gamma^{-1} - 1 - \lambda - La(2\gamma^{-1} - \lambda))\|u_{k+1} - u_k\|^2.
\end{aligned}
$$

---

[2]Strictly speaking we would have to assume that the term before $\|g_k - g_{k-1}\|^2$ is positive, but if it is not we can just discard it and be done with the proof.

The fact that the terms can be telescoped is, with $\alpha = La$ equivalent to

$$2\gamma^{-1} - 1 - \lambda - (2\alpha\gamma^{-1} - \alpha\lambda) \geq -\alpha^2(1 + 2\gamma^{-1}) + \alpha^2\lambda + (2\alpha\gamma^{-1} - \alpha\lambda)$$

which can be simplified to

$$\alpha^2(2 + \gamma - \lambda\gamma) - \alpha(4 - 2\lambda\gamma) + 2 - \gamma - \lambda\gamma \geq 0$$

By solving for $\alpha$ we get the condition

$$\alpha \leq \frac{2 - \lambda\gamma - \gamma}{2 - \lambda\gamma + \gamma}, \tag{16}$$

where from the condition $2\gamma^{-1} - 1 - \lambda - (2\alpha\gamma^{-1} - \alpha\lambda) \geq 0$ we deduce $\alpha \leq \frac{2-\lambda\gamma-\gamma}{2-\lambda\gamma}$, which is redundant in light of (16). The statement follows since we chose $u_0 = u_{-1}$. $\qquad\square$

## A.2 Improved bounds under monotonicity

For the readers convenience we restate the theorem from the main text.

*Theorem 3.2.* Let $F : \mathbb{R}^d \to \mathbb{R}^d$ be monotone and $L$-Lipschitz. If $aL = \frac{2-\gamma}{2+\gamma} - \varepsilon$ for $\varepsilon > 0$ then, the iterates generated by OGDA+ fulfill

$$\frac{1}{k}\sum_{i=0}^{k-1} \|F(u_i)\|^2 \leq \frac{2}{ka^2\gamma^2\varepsilon}\|u_0 + aF(u_0) - u^*\|^2.$$

In particular, we can choose $\gamma = 1$ and $a < \frac{1}{3L}$.

*Proof of Theorem 3.2.* From Theorem A.1 and the fact that $aL = \frac{\gamma-2}{\gamma+2} - \varepsilon$ we need to find an appropriate $\lambda > 0$ such that $aL \leq \frac{2-\lambda\gamma-\gamma}{2-\lambda\gamma+\gamma}$. Given $\varepsilon > 0$ we therefore aim to find a $\lambda > 0$ such that

$$\frac{2-\gamma}{2+\gamma} - \varepsilon \overset{!}{\leq} \frac{2-\lambda\gamma-\gamma}{2-\lambda\gamma+\gamma}. \tag{17}$$

By bringing both terms on one side and the same denominator we obtain the condition

$$\frac{2\lambda\gamma^2}{(\gamma+2)(\gamma+2-\lambda\gamma)} \overset{!}{\leq} \varepsilon. \tag{18}$$

Using the fact that $\lambda \leq 2\gamma^{-1}$ and $\gamma \geq 0$ we can upper bound the left hand side by $\lambda$. Choosing $\lambda = \varepsilon$ therefore ensures that the necessary condition on the step size (16) is satisfied and at the same time yields the dependence on $\varepsilon$ in the denominator of the right hand side in the statement of the theorem.

$\qquad\square$

## A.3 OGDA+ stochastic

For the stochastic analysis we use for convenience the notation $\Delta_{k+1} = \mathbb{E}\big[\|u_{k+1} - u_k\|^2\big]$ and $\tilde{\sigma}^2 = \mathbb{E}\Big[\|\frac{1}{B}\sum_{j=1}^{B}\tilde{F}(u_k, \xi_j) - F(u_k)\|^2\Big]$.

**Lemma A.2.** *Let* $(u_k)$ *be the sequence of iterates generated by stochastic OGDA+, then, for any* $\lambda > 0$

$$L_{k+1} + a\gamma(a - \rho)\mathbb{E}\big[\|g_k\|^2\big] \leq L_k + 2(1+\lambda)\tilde{C}_1\tilde{\sigma} + (1+\lambda^{-1})\tilde{C}_1\Delta_k - \tilde{C}_2\Delta_{k+1}.$$

*where* $\tilde{C}_1 := \max\{0, aL\gamma^{-1} - a^2L^2(1+\gamma^{-1})\}$ *and* $\tilde{C}_2 := \gamma^{-1} - 1 - aL\gamma^{-1}$.

*Proof of Lemma A.2.* From the update of the method we deduce for all $k \geq 0$

$$
\begin{aligned}
\mathbb{E}\big[\|u_{k+1} + a\tilde{g}_k - u^*\|^2\big] &= \|u_k - a\gamma\tilde{g}_k + a\tilde{g}_{k-1} - u^*\|^2 \\
&= \mathbb{E}\big[\|u_k + a\tilde{g}_{k-1} - u^*\|^2 - \langle 2a\tilde{g}_{k-1} - a\gamma\tilde{g}_k, a\gamma\tilde{g}_k\rangle\big] - 2a\mathbb{E}[\langle u_k - u^*, \gamma\tilde{g}_k\rangle] \\
&\leq \mathbb{E}\big[\|u_k + a\tilde{g}_{k-1} - u^*\|^2 + a\gamma\rho\|g_k\|^2 - \langle 2a\tilde{g}_{k-1} - a\gamma\tilde{g}_k, a\gamma\tilde{g}_k\rangle\big],
\end{aligned}
\tag{19}
$$

where we used

$$
2\mathbb{E}[\langle u_k - u^*, \tilde{g}_k\rangle] = 2\mathbb{E}[\mathbb{E}[\langle u_k - u^*, \tilde{g}_k\rangle \,|\, u_k]] = 2\mathbb{E}[\langle u_k - u^*, g_k\rangle] \geq -\rho\mathbb{E}\big[\|g_k\|^2\big]
$$

to deduce the last inequality. It remains to note that

$$
\begin{aligned}
-\langle 2a\tilde{g}_{k-1} - a\gamma\tilde{g}_k, a\gamma\tilde{g}_k\rangle = 4a\gamma^{-1}\langle \tilde{g}_k - \tilde{g}_{k-1}, u_k - u_{k+1}\rangle - (2\gamma^{-1} - 1)\|u_{k+1} - u_k\|^2 \\
- a^2(1 + 2\gamma^{-1})\|\tilde{g}_k - \tilde{g}_{k-1}\|^2,
\end{aligned}
\tag{20}
$$

which follows immediately the way we deduced (10). Now using the definition of $u_{k+1}$ we get

$$
\begin{aligned}
a^2\gamma\mathbb{E}\big[\|g_k\|^2\big] = a^2\gamma\mathbb{E}\big[\|\mathbb{E}[\tilde{g}_k \,|\, u_k]\|^2\big] \leq a^2\gamma\mathbb{E}\big[\|\tilde{g}_k\|^2\big] = \mathbb{E}\big[\gamma^{-1}\|u_{k+1} - u_k\|^2\big] \\
+ \mathbb{E}\big[\gamma^{-1}\|a(\tilde{g}_k - \tilde{g}_{k-1})\|^2 + 2a\gamma^{-1}\langle u_{k+1} - u_k, \tilde{g}_k - \tilde{g}_{k-1}\rangle\big].
\end{aligned}
\tag{21}
$$

Combining (19) to (21) we deduce

$$
\begin{aligned}
\mathbb{E}\big[\|u_{k+1} + a\tilde{g}_k - u^*\|^2\big] + a\gamma(a - \rho)\mathbb{E}\big[\|g_k\|^2\big] \leq \mathbb{E}\big[\|u_k + a\tilde{g}_{k-1} - u^*\|^2\big] - a^2(1 + \gamma^{-1})\mathbb{E}\big[\|\tilde{g}_k - \tilde{g}_{k-1}\|^2\big] \\
- (\gamma^{-1} - 1)\mathbb{E}\big[\|u_{k+1} - u_k\|^2\big] + 2a\gamma^{-1}\mathbb{E}[\langle u_k - u_{k+1}, \tilde{g}_k - \tilde{g}_{k-1}\rangle].
\end{aligned}
$$

We get via Young's inequality

$$
2a\gamma^{-1}\langle u_k - u_{k+1}, \tilde{g}_k - \tilde{g}_{k-1}\rangle \leq a\gamma^{-1}L\|u_k - u_{k+1}\|^2 + L^{-1}a\gamma^{-1}\|\tilde{g}_k - \tilde{g}_{k-1}\|^2.
$$

Combining the previous two inequalities

$$
\begin{aligned}
\mathbb{E}\big[\|u_{k+1} + a\tilde{g}_k - u^*\|^2\big] + a\gamma(a - \rho)\mathbb{E}\big[\|g_k\|^2\big] \leq \mathbb{E}\big[\|u_k + a\tilde{g}_{k-1} - u^*\|^2\big] \\
+ (L^{-1}a\gamma^{-1} - a^2(1 + \gamma^{-1}))\mathbb{E}\big[\|\tilde{g}_k - \tilde{g}_{k-1}\|^2\big] - (\gamma^{-1} - 1 - La\gamma^{-1})\mathbb{E}\big[\|u_{k+1} - u_k\|^2\big].
\end{aligned}
\tag{22}
$$

Next, we need to estimate the difference of the gradient estimators via the difference of the true

$$
\begin{aligned}
\|\tilde{g}_k - \tilde{g}_{k-1}\|^2 &\leq \left(1 + \frac{1}{\lambda}\right)\|g_k - g_{k-1}\|^2 + (1 + \lambda)\|g_k - \tilde{g}_k + \tilde{g}_{k-1} - g_{k-1}\|^2 \\
&\leq \left(1 + \frac{1}{\lambda}\right)L^2\|u_k - u_{k-1}\|^2 + 2(1 + \lambda)\left(\|g_k - \tilde{g}_k\|^2 + \|\tilde{g}_{k-1} - g_{k-1}\|^2\right),
\end{aligned}
$$

where we used the Lipschitz continuity of the operator. Therefore, by taking the expectation, we obtain

$$
\mathbb{E}\big[\|\tilde{g}_k - \tilde{g}_{k-1}\|^2\big] \leq \left(1 + \frac{1}{\lambda}\right)L^2\mathbb{E}\big[\|u_k - u_{k-1}\|^2\big] + 4(1 + \lambda)\tilde{\sigma}^2.
\tag{23}
$$

Plugging (23) into (22) we deduce

$$
\begin{aligned}
\mathbb{E}\big[\|u_{k+1} + ag_k - u^*\|^2\big] + a\gamma(a - \rho)\mathbb{E}\big[\|g_k\|^2\big] \leq \mathbb{E}\big[\|u_k + ag_{k-1} - u^*\|^2\big] + 4(1 + \lambda)\max\{0, L^{-1}a\gamma^{-1} - a^2(1 + \gamma^{-1})\}\tilde{\sigma}^2 \\
+ \max\left\{0, (1 + \lambda^{-1})(La\gamma^{-1} - a^2L^2(1 + \gamma^{-1}))\right\}\Delta_k - (\gamma^{-1} - 1 - aL\gamma^{-1})\Delta_{k+1}.
\end{aligned}
$$

$\square$

*Theorem 3.3.* Let $F : \mathbb{R}^d \to \mathbb{R}^d$ be $L$-Lipschitz satisfying Assumption 1 with $\frac{1}{L} > \rho$ and let $(u_k)_{k\geq 0}$ be the sequence of iterates generated by stochastic OGDA+, with $a$ and $\gamma$ satisfying $\rho < a < \frac{1-\gamma}{1+\gamma}\frac{1}{L}$ then, to visit an $\varepsilon$ stationary point such that $\min_{i=0,\dots,k-1} \mathbb{E}\big[\|F(u_i)\|^2\big] \leq \varepsilon$ we require

$$\mathcal{O}\left(\frac{1}{\varepsilon}\frac{1}{a\gamma(a-\rho)}\mathbb{E}\big[\|u_0 + a\tilde{g}_0 - u^*\|^2\big]\max\left\{1, \frac{4\sigma^2}{aL}\frac{1}{\varepsilon}\right\}\right)$$

calls to the stochastic oracle, with large batch sizes of order $\mathcal{O}(\varepsilon^{-1})$.

*Proof.* Let us first note that the condition $\alpha \leq \frac{1-\gamma}{1+\gamma}$ implies that the positive part in $\tilde{C}_1$ is redundant as the second term in the maximum

$$\alpha\gamma^{-1} - \alpha^2(1 + \gamma^{-1}) > \alpha\left(\gamma^{-1} - \frac{1-\gamma}{1+\gamma}(1 + \gamma^{-1})\right) = \alpha\left(\frac{\gamma^{-1} + 1 - (\gamma^{-1} - \gamma)}{1+\gamma}\right) = \alpha \geq 0,$$

is already nonnegative. Next we remark that the statement

$$\tilde{C}_2 \geq (1 + \lambda^{-1})\tilde{C}_1 \tag{24}$$

for some $\lambda > 0$ is equivalent to $\tilde{C}_2 > \tilde{C}_1$ (with strict inequality). This is, however, precisely the condition of Theorem A.1 but with strict inequality, which is the reason why we require the strict inequality $\alpha < \frac{1-\gamma}{1+\gamma}$ to ensure (24). So we can iteratively apply the statement of Lemma A.2 to deduce

$$a\gamma(a-\rho)\sum_{i=0}^{k-1}\|g_i\|^2 \leq \mathbb{E}\big[\|u_0 + ag_0 - u^*\|^2\big] + 4(1 + \lambda^{-1})\sum_{i=0}^{k-1}\sigma_i^2.$$

We still need to estimate $\lambda^{-1}$ to find the right batch size in order to decrease the last summand to the desired accuracy. By considering (24) we get

$$(1 + \lambda^{-1}) \leq \frac{\tilde{C}_2}{\tilde{C}_1} = \frac{\gamma^{-1} - 1 - \alpha\gamma^{-1}}{\alpha(\gamma^{-1} - \alpha - \alpha\gamma^{-1})} \overset{\alpha \leq 1}{\leq} \frac{\gamma^{-1} - 1 - \alpha\gamma^{-1}}{\alpha(\gamma^{-1} - 1 - \alpha\gamma^{-1})} = \frac{1}{\alpha}$$

By taking $B := \max\{1, \frac{4\sigma^2}{\alpha\varepsilon}\}$ independent samples per iteration, we get the variance

$$\tilde{\sigma}^2 = \frac{\alpha\varepsilon}{4},$$

and thus arrive at a total oracle call complexity as claimed. $\qquad\square$

## A.4 EG+ with adaptive step size

**Lemma A.3.** *Let $F : \mathbb{R}^d \to \mathbb{R}^d$ be $L$-Lipschitz and satisfy Assumption 1. Then, for the iterates generated by Algorithm 3, it holds that*

$$\gamma^{-1}\|\bar{u}_{k+1} - u^*\|^2 \leq \gamma^{-1}\|\bar{u}_k - u^*\|^2 - a_k(a_k\gamma(1-\gamma) - \rho)\|F(u_k)\|^2$$
$$- \left(1 - \frac{\tau a_k}{a_{k+1}}\right)(\|u_k - \bar{u}_k\|^2 + \|u_k - \bar{u}_{k+1}\|^2). \tag{25}$$

*Proof of Lemma A.3.* We start by using Assumption 1 splitting the following term into three

$$\frac{\rho}{2}\|F(u_k)\|^2 \leq \langle a_k F(u_k), u_k - u^*\rangle$$
$$= \langle a_k F(u_k), \bar{u}_{k+1} - u^*\rangle + \langle a_k F(\bar{u}_k), u_k - \bar{u}_{k+1}\rangle + a_k\langle F(u_k) - F(\bar{u}_k), u_k - \bar{u}_{k+1}\rangle. \tag{26}$$

By expressing $F(u_k)$ via the definition of $\bar{u}_{k+1}$ and the three point identity we obtain

$$\langle a_k F(u_k), \bar{u}_{k+1} - u^*\rangle = \gamma^{-1}\langle \bar{u}_k - \bar{u}_{k+1}, \bar{u}_{k+1} - u^*\rangle$$
$$= \frac{\gamma^{-1}}{2}(\|\bar{u}_k - u^*\|^2 - \|\bar{u}_{k+1} - \bar{u}_k\|^2 - \|\bar{u}_{k+1} - u^*\|^2). \tag{27}$$

Similarly, by expressing $F(\bar{u}_k)$ via the definition of $u_k$ we deduce

$$
\begin{aligned}
\langle a_k F(\bar{u}_k), u_k - \bar{u}_{k+1} \rangle &= \langle \bar{u}_k - u_k, u_k - \bar{u}_{k+1} \rangle \\
&= \frac{1}{2} \left( \|\bar{u}_{k+1} - \bar{u}_k\|^2 - \|u_k - \bar{u}_k\|^2 - \|u_k - \bar{u}_{k+1}\|^2 \right).
\end{aligned}
\tag{28}
$$

Lastly, via the Cauchy-Schwarz inequality

$$
\begin{aligned}
a_k \langle F(u_k) - F(\bar{u}_k), u_k - \bar{u}_{k+1} \rangle &\le a_k \|F(u_k) - F(\bar{u}_k)\| \|u_k - \bar{u}_{k+1}\| \\
&\overset{(4)}{\le} \frac{a_k \tau}{a_{k+1}} \|u_k - \bar{u}_k\| \|u_k - \bar{u}_{k+1}\| \\
&\le \frac{a_k \tau}{2 a_{k+1}} (\|u_k - \bar{u}_k\|^2 + \|u_k - \bar{u}_{k+1}\|^2).
\end{aligned}
\tag{29}
$$

Combining (26) to (29) and multiplying by 2 we get

$$
\begin{aligned}
\rho \|F(u_k)\|^2 \le \gamma^{-1} \left( \|\bar{u}_k - u^*\|^2 - \|\bar{u}_{k+1} - u^*\|^2 \right) &- (\gamma^{-1} - 1)\|\bar{u}_{k+1} - \bar{u}_k\|^2 \\
&- \left( 1 - \frac{\tau a_k}{a_{k+1}} \right) \left( \|u_k - \bar{u}_k\|^2 + \|u_k - \bar{u}_{k+1}\|^2 \right).
\end{aligned}
$$

Using the observation that $\gamma a_k F(u_k) = \bar{u}_{k+1} - \bar{u}_k$, gives

$$
\begin{aligned}
\gamma^{-1}\|\bar{u}_{k+1} - u^*\|^2 + a_k(a_k\gamma(1-\gamma) - \rho)\|F(u_k)\|^2 \le \gamma^{-1}\|\bar{u}_k - u^*\|^2 \\
- \left( 1 - \frac{\tau a_k}{a_{k+1}} \right)(\|u_k - \bar{u}_k\|^2 + \|u_k - \bar{u}_{k+1}\|^2).
\end{aligned}
$$

We see that the largest possible range for $\rho$ is achieved for $\gamma = \frac{1}{2}$. □

*Theorem 4.1.* Let $F : \mathbb{R}^d \to \mathbb{R}^d$ be $L$-Lipschitz that satisfies Assumption 1, where $u^*$ denotes any weak Minty solution, with $a_\infty > 2\rho$ and let $(u_k)_{k \ge 0}$ be the iterates generated by Algorithm 3 with $\gamma = \frac{1}{2}$ and $\tau \in (0, 1)$. Then, there exists a $k_0 \in \mathbb{N}$, such that

$$
\min_{i=k_0,\dots,k} \|F(u_k)\|^2 \le \frac{1}{k - k_0} \frac{L}{\tau(\frac{a_\infty}{2} - \rho)} \|\bar{u}_{k_0} - u^*\|^2.
$$

*Proof.* As $a_k$ converges to $a_\infty$, the quotient $a_k/a_{k+1}$ converges to 1. In particular, there exists an index $k_0 \in \mathbb{N}$ such that $a_k/a_{k+1} \le \frac{1}{\tau}$ for all $k \ge k_0$, because $\tau < 1$. We can therefore drop the last term in (25) and sum up to obtain

$$
\|\bar{u}_{k+1} - u^*\|^2 + \sum_{i=k_0}^{k} a_i \left( \frac{a_i}{2} - \rho \right) \|F(u_k)\|^2 \le \|\bar{u}_{k_0} - u^*\|^2.
$$

The desired statement follows by observing that $a_i \ge a_\infty \ge \tau/L$. □

Note that the above proof for the adaptive version of EG+ provides an improvement in the dependence between $\rho$ and $L$ over the analysis of (Diakonikolas et al., 2021) even in the constant step size regime.

# B  Additional statements and proofs

For the sake of completeness we provide a proof of the elementary fact that Minty solutions are a stronger requirement than Stampachia solutions.

**Lemma B.1.** *If $F$ is continuous then every Minty solution is also a Stampacchia solution.*

*Proof.* Let $w^*$ be a solution to the Minty VI and $z = \alpha w^* + (1 - \alpha)u$ for an arbitrary $u \in \mathbb{R}^m$ and $\alpha \in (0, 1)$, then

$$\langle F(\alpha w^* + (1 - \alpha)u), (1 - \alpha)(u - w^*)\rangle \geq 0.$$

This implies that

$$(1 - \alpha)\langle F(\alpha w^* + (1 - \alpha)u), (u - w^*)\rangle \geq 0.$$

By dividing by $(1 - \alpha)$ and then taking the limit $\alpha \to 1$ we obtain that $w^*$ is a solution of the Stampacchia formulation. □

## C  Numerics

For all experiments, if not specified otherwise, we used for OGDA+ and the adaptive version of EG+ the parameter $\gamma = \frac{1}{2}$. For the step size choice of Algorithm 3 we use $\tau = 0.99$. For the CurvatureEG+ method of (Pethick et al., 2022) (with their notation) we use $\delta_k$ equal to $-\rho/2$, where $\rho$ is the weak Minty parameter, if it is known and less than $1/L$; and $-0.499$ times the step size, otherwise. Furthermore we set the parameters of the linesearch to $\tau = 0.9$ and $\nu = 0.99$.

### C.1  Ratio game

The data used to generate the instance displayed in Figure 3 was suggested in (Daskalakis et al., 2020) and is given by

$$R = \begin{pmatrix} -0.6 & -0.3 \\ 0.6 & -0.3 \end{pmatrix}$$

and

$$S = \begin{pmatrix} 0.9 & 0.5 \\ 0.8 & 0.4 \end{pmatrix}.$$

This results in the following objective function for the min-max problem

$$V(x, y) = \frac{-1.2xy + 0.9y - 0.3}{0.4y + 0.1x + 0.4},$$

which gives rise to the optimality conditions

$$-0.12x^* - 0.39x + 0.48 = 0$$

and

$$-0.48y^2 - 0.57y + 0.03 = 0$$

with the solution $(x^*, y^*) = (0.951941, 0.050485)$. For the experiments we used an estimated Lipschitz constant $L = \frac{5}{3}$.

In Figure 5 we can see the reason for the slow convergence behavior of the CurvatureEG+ method observed in Figure 3. Not only is the step size computed by the backtracking procedure smaller than the one chosen by adaptive EG+, see (4), also the second step (update step) uses an even smaller fraction of the already smaller extrapolation step size.

#### C.1.1  Polar Game

For Figure 2 we used the so-called Polar Game introduced in (Pethick et al., 2022) which is given by

$$F(x, y) = (\psi(x, y) - y, \psi(y, x) + x), \tag{30}$$

where $\psi(x, y) = \frac{1}{16}ax(-1 + x^2 + y^2)(-9 + 16x^2 + 16y^2)$ and parameter $a > 0$. In Figure 2 we used $a = \frac{1}{3}$.

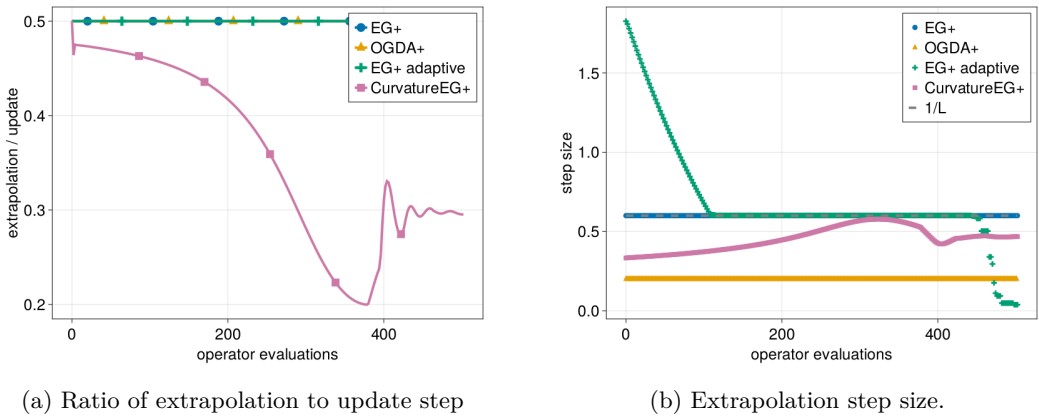

(a) Ratio of extrapolation to update step

(b) Extrapolation step size.

Figure 5: **Ratio game**. An illustration of the step sizes used by different methods, as well as the ratio of extrapolation to update step. CurvatureEG+ chooses its own ratio adaptively and does so for this example in a seemingly overly conservative way, resulting in slow convergence observed in Figure 3.

