# OpenReview forum: "Solving Nonconvex-Nonconcave Min-Max Problems exhibiting Weak Minty Solutions"
_TMLR — Accepted by TMLR_

### Review · Reviewer_aS3x · 2022-12-16

**Summary Of Contributions:**

This paper studies unconstrained min-max problems which have at least one weak-Minty solution. For such problems the authors study an extension of the optimistic gradient descent-ascent algorithm and show that it converges in a slightly larger parameters regime than the extragradient method. The authors also show the same convergence rate as for the latter method. Some other results were also introduced, but I think they were less important.

**Audience:**

Yes

**Broader Impact Concerns:**

...............

**Claims And Evidence:**

Yes

**Requested Changes:**

First of all, I do not understand why the paper's lines are not numbered. I only hope it was not authors' fault. More seriously, here is the list of questions:

1. Abstract. How can we oppose "recently introduce" and "able to capture"? Also   "paragraph" in the end?
2. First paragraph, page 1: "an assumption", not "a".
3. Paragraph "About the weak Minty...", page 2. Any positive number is a multiple of $\rho$. It is not entirely clear to what  $\frac{1}{2L}$ refers: to $\rho$ or to the bound on a step size?
4. page 2: What is "even monotone"? Do we also have "odd monotone"? What does it mean "significantly improved"? From 0.1 to 1000?
5. "Operator norm for monotone problems", page 3. There is no need to mention all non-interesting intermediate results. The goal is to make a paper more readable and not to please all potential reviewers.
6. In general, I feel that  the number of citations is unnecessarily large. A lot of results for which the authors refer are quite trivial. For example, in page 6, the last two display equations are obvious, there is no need to mention another two papers that used them. There are plenty of such cases.
7. Why do we even need to introduce what VI is? We work exclusively with the equation $F(x) = 0$. Why do we need strong monotonicity or cocoercivity? Do we use them in the paper?
8. page 4 (bottom), "solution operator" is not  defined.
9. page 5, why subgradients? We define $F$ to be a single-valued operator.
10. page 5, paragraph after Eq. 2. What is the point of this discussion? What has it to do with the paper's topic?
11. "very ill-behaved"?
12. smooth and $L$-smooth are different classes of functions.
13. Theorem 3.1: Why not to optimize over $\gamma$? This will give a more clear dependence on $\rho$ and $L$.
14. "Effective step size"?
15. Theorem 3.2 is confusing. Isn't $\frac{1}{2L}$ the upper bound for step sizes when $\gamma = 1$? If I am right, I don't think it is that nice to formulate the theorem knowing in advance that with a bit more efforts one can improve it.

16. I would not call Algorithm 2 as adaptive. This is how a term might lose its meaning. In this way, the subgradient method is adaptive or any method with linesearch is adaptive. Adaptivity and not requiring to know $L$ are quite different things. I don't think that a method with decreasing steps should be called adaptive.

    Also, its advantage is slightly overstated. Its theoretical convergence requires $a_{\infty}> 2\rho$ which in turn require us to know $L$.


**Strengths And Weaknesses:**

In my opinion, it is far-fetched to motivate such problems by GANs or adversarial networks. At the same time, there are few other classes of operators that have clear potential. For such classes of operators, the paper does provide a meaningful result.  However, the paper writing has to be improved quite significantly.

---

> ### Author Response · Authors · 2023-01-03
> **Response**
>
> First of all we would like to thank the reviewer for the comments that will surely improve the writing of the paper.
>
> ad "No line numbering") We double checked and are using the version of the tmlr latex package (i.e. no optional arguments) suggested in the template. However, we agree that these are suboptimal circumstances.
>
> ad 1) Replaced "but" by "and" and slightly reformulated the sentence.
>
> ad 2) Done.
>
> ad 3) Fixed the entire paragraph (see requests 2. and 3. of reviewer LpsF)
>
> ad 4) Fixed to "moreover".
>
> ad 5) Our intent was not to please reviewers, but we agree that the paragraph didn't contribute enough to the quality of the paper to justify its existence so we removed it.
>
> ad 6) We streamlined the two display equation and decluttered some of the less relevant references.
>
> ad 7) In our opinion it makes sense to introduce VIs as the (weak) Minty VI requires the variational (inner product) formulation even without the presence of constraints. This way it is easier for the reader to see the difference.
> While not strictly necessary we feel it makes sense to recall the definition of cocoercivity as it is relevant to cohypomonotonicity (only the sign is changed), which in turn is relevant to the weak Minty notion used in the paper. However, we acknowledge the criticism and made the paragraph more terse in order to not distract too much.
>
> ad 8) Fixed.
>
> ad 9) Removed the "sub" in "sub-gradient".
>
> ad 10) We made the paragraph more terse.
>
> ad 11) Replaced by "erratic" and explained what we mean, i.e. limit cycles and spurious attractors.
>
> ad 12) We did not want to introduce the definition of $L$-smoothness and thus stuck to "smooth" and referred to the Lipschitz continuity of the gradient explicitly when used. At one point this distinction was not clear, where we fixed it.
>
> ad 13) Due to the step size appearing squared in the denominator and the step size upper bound being a fraction in gamma this unfortunately gives an uninsightful on rho. We also want to highlight the fact that the convergence statement is not so much about the precise constant but more about the fact that can solve problems up to $\rho < \frac{1}{L}$ at all.
>
> ad 14) Removed the word "effective".
>
> ad 15) It is true that the $\frac{1}{2L}$ upper bound for the step size is sufficient to obtain convergence of the iterates and the gap function. For rates for the norm of the operator, however, $\frac{1}{3L}$ is actually the best known one, please see the discussion we added after the theorem.
>
> ad 16) We agree that there exist different levels of adaptivity that vary in their usefulness and that the one we propose might not be the most powerful one. Would the reviewer also argue that AdaGrad is not adaptive because its step size is nonincreasing? We only mean by adaptive that no step size needs to be supplied by the user and no linesearch is carried out. We added this explanation to the first paragraph of section 4.

---

### Review · Reviewer_LpsF · 2022-12-19

**Summary Of Contributions:**

The work focuses on weak Minty solutions of Lipschitz operators. The authors consider three methods: Optimistic Gradient Descent Ascent (OGDA) also known as Popov's method, its stochastic version, and the adaptive version of the celebrated Extragradient method (EG). The convergence is proven with respect to the averaged squared norm of the operator. The authors also provide several numerical experiments showing that OGDA can converge for some weak Minty problems when EG fails and illustrate the advantage of using the new version of adaptive EG.

**Audience:**

Yes

**Broader Impact Concerns:**

I have no broader impact concerns.

**Claims And Evidence:**

Yes

**Requested Changes:**

## Requests

1. If I am not mistaken, the analysis relies on the fact that the solution is unique. I suggest adding this clarification to the paper. Can the analysis be generalized to cover the case when there are multiple solutions (e.g., when (1) holds only for the closest solution to the considered point $u$)?

2. Page 2, "it is necessary that step size is larger than a multiple of $\rho$": could the authors provide references supporting this claim?

3. Page 2, the same paragraph: the result of Pethick et al. (2022) holds for any $\rho < 1/L$.

4. After inequality (2), "they appear as gradients of smooth convex functions": I think it is important to mention that cocoercive operators are not necessary gradients of smooth convex functions (e.g., rotation operators with acute angles). The current formulation can confuse a reader.

5. Section 3.2 requires some formalization: I suggest adding the pseudocode of the method and explicitly showing that it uses mini-batches.

6. Could the authors provide a complete derivation of the first formula in the proof of Theorem 3.2?

7. At the end of the proof of Theorem 3.2, the authors refer to the lemma of Opial. Can the authors provide a reference and complete formulation in the appendix?

8. Can the authors elaborate on how the last inequality from the proof of Lemma A.2 implies the statement of the lemma? For example, I believe one should mention that $a \leq L\gamma$ to get this result and also write $\max(0, \ldots)$ in front of $\sigma_{k-1}^2$ in the last inequality from the proof (similarly to the factor in front of $\Delta_k$).

9. Proof of Theorem 3.3. I request the authors to provide a complete derivation and formalize a bit the proof. In particular, the derivation of $\tilde{C}_2 \geq \tilde{C}_1$ is missing, instead of $||u_1 - u_0||^2$ one should have $\mathbb{E}||u_1 - u_0||^2$, and in (22) it should be somehow reflected through the notation that summands denote different samples.

## Suggestions, comments and questions

1. How is Theorem 3.2 related to the existing results for OGDA?

2. Is there a way to justify the necessity of large batch sizes in Theorem 3.3?

3. Why are $u_{k-1}$ and $\bar{u}_{k-1}$ used in line 2 of Algorithm 2 to estimate the Lipschitz constant? From my understanding, it mostly comes from the analysis, but it would be good to understand whether there is another reason explaining why we should use this pair of points, e.g., why

$\bar{u}_k, u_{k-1}$

or

$\bar{u}_k, \bar{u}_{k-1}$

are not used instead? Some numerical comparisons of different modifications can also be very interesting to see.

4. I suggest repeating the formulations of the theorems in the appendix right before their proofs for the readers' convenience (e.g., in Appendix A.2).

## Minor comments

1. The last word of the abstract ("paragraph") should be removed.

2. Theorem 3.1, last sentence: "$\gamma$ small enough" $\to$ "small enough $\gamma$".

3. Theorem A.1, "Let ... and $(u_k)_{k\geq 0}$ {\color{red} be} the sequence": here "{\color{red} be}" is missing.

4. The authors missed periods after some formulas, e.g., (15), formula between (19) and (20), (20).

5. Lemma A.2, the second term in the left-hand side: factor $\gamma$ is missing. Moreover, $L_k$, $\sigma_k$ and $\tilde g_k$ are undefined.

6. Formula after (19): expectation is missing in the right-hand side.

7. The last step of the proof of Lemma A.3: it is better to write a complete derivation (though it is correct and can be verified). Also, it is better to explain why the choice of $\gamma = 1/2$ is the best and in what sense. I guess this is because it makes the factor in front of $||F(u_k)||^2$ as large as possible in the proof. However, there are other terms dependent on $\gamma$ as well.

**Strengths And Weaknesses:**

## Strengths

1. All the results are correct and new. I have checked the proofs in detail and found only minor inaccuracies. Overall, the paper is well-written as well.

2. For OGDA+ the authors derive $O(1/k)$ convergence rate for any $\rho < 1/L$ (here $\rho$ is the weak Minty constant and $L$ is Lipschitz constant). This matches the best-known results for EG obtained by Pethick et al. (2022). It is worth mentioning that the proposed analysis differs from the existing ones known for OGDA and EG.

3. The authors propose a version of EG+ with adaptive step size and derive $O(1/k)$ convergence rate for the proposed algorithm for finding weak Minty solutions. The proposed method shows good performance in the experiments as well.

## Weaknesses

In my opinion, weaknesses are minor and are outweighed by strengths.

1. (minor) The stochastic version of OGDA is analyzed only in the case of large $O(\varepsilon^{-2})$ batch sizes. Although it is a known problem of deriving $O(1/k)$ convergence rates in terms of the (expected) squared operator norm without using large batch sizes for stochastic EG and OGDA (even in the monotone case), it would be good for the paper to add this discussion somewhere in the text, e.g., right after Theorem 3.3.

2. (minor) The rate for Theorem 4.1 is derived only for $k > k_0$, where $k_0$ is unknown in general. Is it possible to estimate $k_0$ somehow at least under some additional assumption? It would be interesting to have some upper bound on $k_0$ in some special cases (e.g., for monotone problems).

---

> ### Author Response · Authors · 2023-01-03
> **Response**
>
> First of all we would like to especially thank reviewer LpsF for their thorough read of our
> manuscript and the insightful comments.
>
> **Weaknesses**
>
> ad 1) We added a discussion about how small/decreasing learning rates are in
> practice preferred in practice and how this is difficult for problems with weak
> Minty solutions, as they generally rely large step sizes.
>
> ad 2) We added a clarifying comment that this problem cannot be mitigated even
> in the monotone setting where author have focused on the convergence of the
> iterates -- without any rate.
>
> **Requested changes**
>
> ad 1) In fact then analysis does not require a unique solution. We added a
> clarifying "$u^*$ denotes any weak Minty solution" in the statement of the
> theorems.
>
> ad 2&3) We clarified that what we meant was rather "larger than a term
> proportional to $\rho$". We also reworded the entire paragraph for the sake of
> clarity.
>
> ad 4) Thank you. We added a "-- for example --".
>
> ad 5&6) Done.
>
> ad 7) We removed the reference to Opials lemma and instead reformulated the
> theorem to be only in terms of operator norm to be more consistent with the
> other theorems. This does not require Opials lemma (it was only needed for the
> convergence of the iterates).
>
> ad 8&9) We would like to again thank the reviewer for their careful read - apparently also of the proofs. We indeed missed a term here which is now corrected and results in a slightly modified batch size.
> We also simplified the entire proof to make it more readable and were able to improve the constants. The step size is now only constrained by $1/L$ as in the deterministic case. This is due to the fact that the constants $\tilde{C}_1$ and $\tilde{C}_2$ are now identical to the respective terms in the deterministic proof. Therefore also the desired inequality follows trivially. Furthermore, the term $\Vert u_1-u_0 \Vert^2$ was a result expanding the recursion in iteration to few, which is now fixed.
>
> **Suggestions, comments and questions**
>
> ad 1) We added a paragraph highlighting that for $\gamma=1$ (i.e. regular OGDA) we match the best known (and very recent) step size bound for rates on the operator norm, but improve the bounds in the generalized case.
>
> ad 2) Please see our reply to 1) in *weaknesses*.
>
> ad 3) That's a very interesting question, especially considering that when computing the step size already $ \bar{u_k} $ is available, but we are still using $\bar{u}_{k-1}$. We ran some experiments, but saw almost no difference (the current version was actually slightly faster on some instances). We also went back and checked the references in the literature that use a similar step size and found they that they also don't use the very latest iterate. The insight that the current form comes from the analysis is of course correct, but that does not mean that a different version is not possible. We are afraid that a further investigation will be beyond the scope of this revision, but we will keep it in mind for future work.
>
> ad 4) Done.
>
> **Minor comments**
>
> ad 1&2&3&4&5&6) Done.
>
> ad 7) We added an explanation that we mean that the choice $\gamma=1/2$ is optimal in the sense that it allows for the largest range of $\rho$.

---

> > ### Comment · Reviewer_LpsF · 2023-01-15
> > **Reply to authors**
> >
> > I thank the authors for the updates and replies to my comments. All requests are properly handled by the authors.
> >
> > I have only a minor remark about the solution set. I agree that the current formulation of Assumption 1 and the analysis allow considering any solution $u^\ast$. This is achieved via assuming that $u^\ast$ is such that *for any* $u$ inequality (1) holds. However, similar conditions (quasi-strong monotonicity, quadratic gradient growth, quadratic functional growth -- see more in [Necoara, I., Nesterov, Y., & Glineur, F. (2019). Linear convergence of first order methods for non-strongly convex optimization. Mathematical Programming, 175(1), 69-107]) are usually assumed to hold only for a projection of $u$ on the solution set. So, I am curious whether it is possible to generalize the analysis in this paper to the case when Weak Minty solutions are defined slightly differently: $U^\ast$ is a set of Weak Minty solutions if for any $u \in \mathbb{R}^d$ and $u^\ast = \text{proj}_{U^\ast}(u)$ (projection of $u$ on the solution set) inequality (1) holds.

---

> > > ### Author Response · Authors · 2023-01-19
> > > **reply about generalizing weak Minty**
> > >
> > > That is an interesting perspective, so we are thankful for the comment. However, currently we don't see a way to apply this here. First of all we don't know whether weak Minty solutions are necessarily stationary points. So it is not clear which set to project on (only if we knew that every weak Minty solution is also a stationary point we can hope to find a generalization). So for the sake of the argument let's assume this to be the case. If $u^*$ in the definition of a weak Minty is replaced by the projection onto the set of stationary points - since no $|| u -u^* ||^2$ term shows up in the definition - it is not clear why the condition would be relaxed by taking a closer point.
> > > To put it differently, if the definition of weak Minty would ask for inequality (1) to hold for any stationary point, we could weaken it by instead requiring it only to hold for a specific one (the projection). Since this is not the case, the benefit is, at least to us, not obvious.

---

### Review · Reviewer_QNP8 · 2022-12-27

**Summary Of Contributions:**

This paper considers the optimistic gradient descent ascent (OGDA) for a class of structured variational inequalities problems and its applications to the min-max problem. The major difference lies in the weak Minty assumption. The authors present the convergence of the proposed algorithm in both deterministic and stochastic cases. Under the stronger assumption that the operator is even monotone, the authors show that the algorithm could use large step sizes.

**Audience:**

Yes

**Broader Impact Concerns:**

I do not find any ethical concern.

**Claims And Evidence:**

Yes

**Requested Changes:**

1. The authors need to explain the necessity of studying the weak Minty solution problem. More specifically, they need to provide sufficient conditions to check the weak Minty property. What kinds of machine learning problems enjoy such a property?

2. They need to explain the difference between cohypomonotonicity and weak Minty property, that is, at least providing some examples that are weak Minty but not cohypomonotonic.

**Strengths And Weaknesses:**

 Strengths： This paper brings some new insights for solving variational inequalities. The authors prove the complete convergence and theoretical advantage of their proposed algorithms.

However, some weaknesses prevent me from voting acceptance of this paper.

1. The weak Minty solution is a direct weak version of cohypomonotonicity [Bauschke et al. (2020); Combettes & Pennanen (2004)], which greatly weakens the novelty.

2. The authors fail to explain the necessity of studying the weak Minty solution problem. What kinds of machine learning problems enjoy such a property? Why are we interested in this property?

---

> ### Author Response · Authors · 2023-01-03
> **Response**
>
> We would like to thank the reviewer for their time and hope to mitigate some of their concerns.
>
> - *"The weak Minty solution is a direct weak version of cohypomonotonicity, which greatly weakens the novelty. [...] They need to explain the difference between cohypomonotonicity and weak Minty property."*
>
>   Even in the cohypomonotone setting, our results on OGDA are novel. In general both settings are studied much less (from an algorithmic standpoint) compared to monotone problems and are therefore arguably novel.
>
>   One could argue that probably most problems with weak Minty solution are also cohypomonotone, but(!) with a different parameter. And therein lies the crux. For example, we added a new reference showing monotonically decreasing operator norm for the extragradient methods when applied to cohypomonotone problems. This stands in stark contrast the what we observe from the experiments in our manuscript highlighting the relevance in distinguishing the two settings.
>   We added a sentence in the paragraph about cohypomonotonicity in the manuscript about this observation.
>
> - *"What kinds of machine learning problems enjoy such a property? Why are we interested in this property?"*
>
>   As we explained for example in the paragraph "Leaving the monotone world." General nonconvex-nonconcave min-max problems (which most of the applications listed in the introduction fall into) can be quite erratic and we are not aware of any global convergence results of first order methods. We therefore argue that it makes sense to study problem classes that go beyond monotonicity -- especially if they show interesting behavior not present for monotone problems, even though they might not be able to capture the true problem always.
>
> - *"More specifically, they need to provide sufficient conditions to check the weak Minty property."*
>
>   There is no way to check. Nevertheless, the insight gained in this paper can still be used to see which modifications need to be made to the popular OGDA method to make it better suited to at least some non-monotone VIs (the ones with weak Minty solution).

---

### Review · Reviewer_dg7N · 2023-01-06

**Summary Of Contributions:**

This paper studies the problem of finding a zero of an operator that is negatively cocoercive wrt a solution $u^*$. In other words, there exists a weak Minty solution of the problem $u^*$.

The main contribution is to develop a modification of OGDA called OGDA+ which provably converges at the rate $1/k$ in terms of the squared operator norm. In comparision, Diakonikolas et al 2021 recently developed a modification of the Extra Gradient algorithm called EG+ that matches the rate $1/k$. It was not known if such a modification could be applied to OGDA which is a concurrent algorithm to EG.

The authors also develop an adaptive version of EG+ which does not require the knowledge of the Lipschitz constant. However, the established convergence rate holds after an unknown number $k_0$ of steps.






**Audience:**

Yes

**Claims And Evidence:**

Yes

**Requested Changes:**



Paragraph after Th 3.1. Given a step size $a$, what is the optimal value of $\gamma$ ?

As I said, the preliminaries are well written but I have a small question:
monotonicity implies star-monotonicity, however, negative cocoercivity does not imply negative star cocoercivity (because $F(u^*)$ might not be equal to zero), am I correct? So, negative star cocoercivity is not a generalization of negative cocoercivity.

In general, it is not clear to me that the operator is zero at a weak minty solution, i.e. $A(u_k) \to A(u^*) = 0$ (which should be true since you look at the squared operator norm).

First sentence after Th 3.3. What is the connection between large batch size and large step size?
One call to the stochastic oracle is computing $g_k$ or $F(u,\xi)$?





**Typos**

"we have to decrease $\gamma$ as evident from (10)". From (3) I guess?


" more then ten times"

 "problem (30) displayed in Figure 1." In general, check the numbering of the equations.

**Strengths And Weaknesses:**

The paper is very well written, and the presentation is very clear. The introduction and the preliminaries clearly explain the problem, the concept of weak Minty and its connection to monotonicity, and the contribution in light of previous works (which I summarized in the previous paragraph). I appreciated to read this part and I believe that it could even be the beginning of a review paper on the topic.

The theoretical claims are cast in theorems (summarized above) proven in the appendix. The only issue I see is that the convergence rate of the adaptive algorithm holds after an unknown number of iterations $k_0$. The proofs are rather standard optimization proofs, but clearly presented.

Finally, some honest simulations show the advantage of each proposed algorithm: the advantage of the adaptive method over the others on a toy problem exhibiting a repelling cycle around the solution, and a comparison of EG+ and OGDA+ (proposed algo) where, on some problem OGDA+ is better, and on some difficult instance of a min max problem from (Daskalakis et al. (2020);
Diakonikolas et al. (2021)) EG+ is slightly better.

---

> ### Author Response · Authors · 2023-01-10
> **Response**
>
> We would like to thank the reviewer for their time, their constructive feedback and encouraging words.
>
> - **"monotonicity implies star-monotonicity, however, negative cocoercivity does not imply negative star cocoercivity (because $F(u^*)$ might not be equal to zero), am I correct? So, negative star cocoercivity is not a generalization of negative cocoercivity."**
>
>     Yes, there are some nuances here. First of all, in any case, one has the assume the existence of a stationary point (i.e. $\exists u^*$ such that $F(u^*)=0$), which we briefly mentioned in the paragraph about negative comonotonicity. Secondly, assuming that this is the question here, in the presence of constraints (which is not the case in the paper) one would have to adapt the definition to incorporate the normal cone, i.e. if the VI is given by $T := F + N_C$, where $N_C$ is the normal cone with respect to some closed and convex set $C$, then one would need to ask the operator $T$ to be negative cocoercive. Then, in turn $T(u^*)$ would naturally be equal to zero and the stated implication holds (if a solution exists).
>     Either way, we added a sentence in section 2 emphasizing again the need for the existence of a stationary point to be able to conclude the weak Minty property from neg. cocoercivity.
>
> - **"In general, it is not clear to me that the operator is zero at a weak minty solution, i.e. $A(u_k) \to A(u^*) = 0$ (which should be true since you look at the squared operator norm)."**
>
>     At this point it is not clear whether a weak Minty solution is also a stationary point $F(u^*)=0$, although we suspect so, but did not want to speculate in the paper. We made no statements in the paper about convergence of the iterates, but it only takes a few more arguments to conclude the boundedness of the iterates and thus conclude convergence of a subsequence to a point $u'$ such that $F(u')=0$. We note, however, that such a point is not necessary a weak Minty solution. A similar observation was remarked in Diakonikolas et al. (2021).
>
> - *"Paragraph after Th 3.1. Given a step size $a$, what is the optimal value of $\gamma$ ?"*
>
>     There is no simple and satisfying answer here.
>     Given the step size, we would like $\gamma$ as large as possible such that $aL \le (1-\gamma)/(1+\gamma)$. However, it is probably not a realistic setting that the step size is known and only $\gamma$ has to be tuned. If $\rho$ and $L$ are known, due to the fractional dependence of $\gamma$ the step size, we would get an uninsightful cubic dependence of $\gamma$ on $\rho$. We added a sentence at the end of the mentioned paragraph about how to choose a practical $\gamma$: small enough to get convergence but not smaller. (This assumes an estimate of $L$.) Last but not least, we also want to highlight the fact that the convergence statement is not so much about the precise constant but more about the fact that can solve problems up to $\rho<1/L$ at all.
>
>
> - *"What is the connection between large batch size and large step size?"*
>
>     To the best of our knowledge there exist only a limited set of strategies to guarantee convergence of stochastic algorithm: small/decreasing step sizes, large/increasing batch sizes; or variance reduction.
>     We tried to explain how the first one of these strategies causes problem weak Minty problems so we opted for the second one (the last one is out of scope). We simplified the corresponding paragraph in the hopes of making this point clearer.
>
> - *"One call to the stochastic oracle is computing $g_k$ or $F(u,\xi)$?"*
>
>     Fixed.
>
> - *"problem (30) displayed in Figure 1."*
>
>     The numbering was correct but the definition of the problem was in the appendix. We moved it to the main text - which we should have done right away - as it is the one problem where OGDA+ is the only method able to converge.

---

> > ### Comment · Reviewer_dg7N · 2023-01-18
> > **Small clarification**
> >
> > Thanks, I am satisfied with the answers, I just need one clarification.
> >
> > The weak minty assumption is just an assumption about the existence of a distinguished point, but then the algorithm does not have to converge to it. Besides, your paper says that the algorithm does drive F(u^k) to zero, under the existence of this distinguished point. Is that correct?

---

> > > ### Author Response · Authors · 2023-01-19
> > > **clarification**
> > >
> > > Exactly.

---

### Comment · Action_Editors · 2023-01-29
**Final Decision**

Dear authors,

The reviewers agree that this is a good paper the community will benefit from. The paper tackles an interesting problem and is very well written. I agree with the recommendation of the reviewers, and recommend acceptance. Please apply the most reasonable changes requested by the reviewers for the camera-ready version of the paper. Most of them are very minor.

Congratulations and thanks for submitting your work to TMLR!

Bets regards,

Action Editor
TMLR

---

### Author Response · Authors · 2023-02-08
**Camera-ready version**

Dear action editor and reviewers,
we are glad that that the paper has been accepted and would like everybody for their effort in improving it. We have now uploaded the camera-ready version.

---

### Decision · Action_Editors · 2023-02-25

**Recommendation:** Accept with minor revision

**Comment:**

Dear authors,

The reviewers agree that this is a good paper the community will benefit from. The paper tackles an interesting problem and is very well written. I agree with the recommendation of the reviewers, and recommend acceptance. Please apply the most reasonable changes requested by the reviewers for the camera-ready version of the paper. Most of them are very minor.

Congratulations and thanks for submitting your work to TMLR!

Bets regards,

Action Editor TMLR

**Audience:**

The paper will certainly be of interest to theoreticians working in the area of min-max problems.

**Claims And Evidence:**

Yes, all claims are supported with satisfactory evidence:
a) This is a theoretical paper - the key contributions are theorems, complete with proofs.
b) Moreover, suitably selected and well executed computational experiments complement the proofs.